

# Projective cluster-additive transformation for quantum lattice models

**Max Hörmann[*] and Kai P. Schmidt[†]**

Department of Physics, Friedrich-Alexander-Universität Erlangen-Nürnberg (FAU),
Staudtstraße 7, Germany

[*] max.hoermann@fau.de ,    [†] kai.phillip.schmidt@fau.de

## Abstract

We construct a projection-based cluster-additive transformation that block-diagonalizes wide classes of lattice Hamiltonians $\mathcal{H} = \mathcal{H}_0 + V$. Its cluster additivity is an essential ingredient to set up perturbative or non-perturbative linked-cluster expansions for degenerate excitation subspaces of $\mathcal{H}_0$. Our transformation generalizes the minimal transformation known amongst others under the names Takahashi's transformation, Schrieffer-Wolff transformation, des Cloiseaux effective Hamiltonian, canonical van Vleck effective Hamiltonian or two-block orthogonalization method. The effective cluster-additive Hamiltonian and the transformation for a given subspace of $\mathcal{H}$, that is adiabatically connected to the eigenspace of $\mathcal{H}_0$ with eigenvalue $e_0^n$, solely depends on the eigenspaces of $\mathcal{H}$ connected to $e_0^m$ with $e_0^m \leq e_0^n$. In contrast, other cluster-additive transformations like the multi-block orthogonalization method or perturbative continuous unitary transformations need a larger basis. This can be exploited to implement the transformation efficiently both perturbatively and non-perturbatively. As a benchmark, we perform perturbative and non-perturbative linked-cluster expansions in the low-field ordered phase of the transverse-field Ising model on the square lattice for single spin-flips and two spin-flip bound-states.



# 1   Introduction

In order to solve the time-independent Schrödinger equation for a Hamiltonian on a lattice

$$\mathcal{H} = \mathcal{H}_0 + \lambda V \,, \tag{1}$$

one needs to find the eigenvalues and eigenfunctions of $\mathcal{H}$. We will assume throughout that $\mathcal{H}_0$ is solvable and has a gapped spectrum. The part $\mathcal{H}_0$ can therefore be written in diagonal form, while

$$[\mathcal{H}_0, V] \neq 0 \tag{2}$$

makes solving $\mathcal{H}$ a difficult problem. Many times one is not interested in properties at all energies of the many-body Hamiltonian but only in the properties of the ground-state and a few low-lying excitations and thus in much fewer degrees of freedom. Conceptionally, one can try to find a transformation $T$ that maps the full Hamiltonian to an effective Hamiltonian $\mathcal{H}_{\text{eff}}$ describing these relevant degrees of freedom only. In practice, in almost all cases one can not find this transformation exactly but has to resort to approximations. One of the oldest is perturbation theory. Let us note that the necessity of a perturbative starting point is not only a drawback but also helps in giving a clear picture of the physical problem at hand. While the first two orders of perturbation theory normally can be easily calculated by hand, high orders are only accessible with computer aid and several methods for their computation exist. Albeit many other numerical techniques exist nowadays, high-order series expansions are used as a competitive technique to tackle quantum many-body problems at zero temperature [1–3]. Examples range from the calculation of low- and high-field expansions for transverse-field Ising models [4,5], the analysis of phase transitions in triangular-lattice bilayer Heisenberg models [6] and spectral densities of two-particle excitations in dimerized Heisenberg quantum spin systems [2,7,8] to the study of critical and Griffiths-McCoy singularities in quantum Ising spin-glasses [9] or the derivation of spectral densities for Heisenberg quantum magnets with quenched disorder [10,11], or to the analysis of quantum phase diagrams of long-range transverse-field Ising models [12] and the application to quantum phases with intrinsic topological order [13–15]. Also questions such as the exploration of possible ground states in the kagome Heisenberg model [16] can be tackled with perturbation theory. In all these examples, the quantum phase transitions are investigated by applying extrapolation techniques to high-order series expansions of relevant energies or observables to investigate the breakdown of the quantum phase present at $\lambda = 0$. The accuracy of those increases with higher orders of perturbation available. This shows that the efficiency of the method used to derive the perturbative expansion is crucial.

A common approach to calculate quantities perturbatively on a lattice is to do a graph decomposition. Especially in dimensions larger than one, this becomes essential for obtaining

high orders. Instead of a large single cluster, the calculations are performed on many small ones, which decreases memory requirements and is easily parallelized. The calculated values of a quantity $M$ on the subgraphs of the lattice are then multiplied with embedding factors to obtain the value of $M$ up to a given order on the whole lattice making use of the inclusion-exclusion principle. If for two disconnected parts $A$ and $B$ of the lattice, the operator $M(A \cup B)$ is the direct sum

$$M(A \cup B) = M(A) \oplus M(B), \tag{3}$$

the graph expansion can be restricted to connected subgraphs of the lattice. An operator $M$ that fulfils property (3) is called additive. However, not every transformation yields an effective Hamiltonian that allows a decomposition of the form (3). In particular, the efficient block-diagonalisation transformation, that only makes use of the projectors of eigenspaces of $\mathcal{H}_0$ and $\mathcal{H}$ (see next section for a detailed introduction), in general, does not allow to perform calculations on linked subgraphs of the lattice only. This is unfortunate since it can be efficiently calculated using matrix-vector multiplications only [3]. This transformation was introduced by different people in different communities. Because of that it is known under different names, for example as Takahashi's transformation, Schrieffer-Wolff transformation, des Cloiseaux effective Hamiltonian, canonical van Vleck effective Hamiltonian or two-block orthogonalization method [3, 17–20]. The existence of many different formulations of the same transformation demonstrates its generic relevance but it is partially surprising that connections between formulations are not well documented.

An obvious drawback of perturbative results is the limitation to the convergence radius of the perturbative expansion. This radius can often be extended significantly by extrapolations. Even though for many models extrapolations are very helpful in determining phase boundaries or critical behaviour, there are some where no conclusive answer can be reached. Another solution to extend beyond the convergence radius of the perturbative expansion are non-perturbative linked-cluster expansions (NLCEs). First introduced in [21], they were often used for thermodynamic quantities [22] or ground-state expectation values [23]. In contrast to quantum Monte Carlo simulations, frustration poses no technical problem. NLCEs also do not suffer from high dimensions as density-matrix renormalization group does. The same holds true for the perturbative linked-cluster expansions. NLCEs follow the same principles as perturbative expansions but use non-perturbative cluster results, which are in many cases just the exact results of the finite cluster. They are again only expected to converge within the quantum phase adiabatically connected to the limit $\lambda = 0$. However, there is hope that NLCEs are helpful for models where perturbative series extrapolations fail. NLCEs have the potential to converge whenever a finite correlation length is present and to allow for scaling close to critical points.

For non-perturbative expansions it is even more important that the expansion can be performed on linked clusters only. Otherwise finding a hierarchy to truncate the expansion is difficult. For excited states non-perturbative linked-cluster expansions were performed with flow-equations in an approach called graph-based continuous unitary transformations (gCUT) [24]. Another expansion, only relying on the eigenvectors and energies of the block of interest, is the contractor renormalization group method (CORE) [25]. In contrast to gCUT, it does not fulfil the linked-cluster property in general. However, a great advantage is its efficiency only relying on the low-energy eigenstates that can be calculated with numerical routines such as the Lanczos algorithm. The CORE method is therefore similar to the projective transformation mentioned above. Although an implementation is as straightforward as for the CORE approach, no NLCEs using the projective transformation are known to us.

Altogether, the projective transformation has therefore many benefits but a crucial drawback: for multi-particle excitations in general no linked-cluster expansion is possible. This restricts the applicability to a limited number of models and forces one to use less efficient methods.

So far, the non-validity of a linked-cluster expansion for this transformation is not well understood. In this paper, we will identify the origin of the problem and will introduce an optimal modified projective transformation, where this problem is absent. We do this by extending the projective transformation for an eigenspace adiabatically connected to $e_0^n$, where $e_0^n$ denotes the energy of the degenerate subspaces of $\mathcal{H}_0$, to incorporate eigenstates adiabatically connected to blocks $m$ with $e_0^m < e_0^n$ and not only those of $e_0^n$. This method shares the efficiency of the projective method, can be applied non-perturbatively using the exact lowest eigenvectors and energies, and allows for cluster expansions with linked clusters only.

Before describing the important changes to the transformation we review other approaches to construct a genuine linked-cluster transformation and inform about different equivalent formulations of the classical projective transformation in Sec. 2. Then we exemplify the roots of the linked-cluster violation of the projective transformation with a simple toy model. In Sec. 3 we show how these problems can be cured for multi-particle excitations in general and also give a general form of the transformation in terms of projection operators. As an application, in Sec. 4 we apply the method to the low-field expansion of the TFIM on the square lattice, both perturbatively and non-perturbatively. We conclude our work in Sec. 5.

## 2 Block-diagonalisation methods

In this section, we first define what block-diagonal form we want to achieve with block-diagonalisation methods and fix basic notation. Then we review existing cluster-additive block-diagonalisation methods and the projective minimal transformation.

### 2.1 Block-diagonalised form and cluster-additivity

The Hilbert space $\mathcal{H}$ of a Hamiltonian with local Hilbert space dimension $a$ and $N$ sites has finite dimension $a^N$ and can be written as the direct sum of the eigenspaces $\mathcal{H}_0^n$ of the operator $\mathcal{H}_0$:

$$\mathcal{H} = \bigoplus_{n=0}^{N} \mathcal{H}_0^n \,. \tag{4}$$

As $\mathcal{H}_0$ is assumed to have block diagonal form we have

$$\mathcal{H}_0 = \bigoplus_{n=0}^{N} \mathcal{H}_0^n \,, \tag{5}$$

where the ordering of eigenvalues of the eigenspaces is $e_0^m \leq e_0^n$ for $m \leq n$.

In more explicit form the parts $\mathcal{H}_0^n$ fulfil

$$\mathcal{H}_0 v = \left( \bigoplus_{n=0}^{N} \mathcal{H}_0^n \right) v = \left( \bigoplus_{n=0}^{N} \mathcal{H}_0^n v_{0,n} \right) \,, \tag{6}$$

for $v = \sum_{n=0}^{N} v_{0,n}$ and $v_{0,n} \in \mathcal{H}_0^n$.

For a block-diagonalising unitary transformation $T$ and the corresponding effective Hamiltonian $\mathcal{H}_{\mathrm{eff}} = T^\dagger \mathcal{H} T$, unitarity implies

$$\mathcal{H} = \bigoplus_{n=0}^{N} \mathcal{H}_{\mathrm{eff}}^n = \bigoplus_{n=0}^{N} T \mathcal{H}_0^n \,, \tag{7}$$

as well as $\mathcal{H}_{\mathrm{eff}}$ to be block-diagonal so that it can be written as

$$\mathcal{H}_{\mathrm{eff}} = \bigoplus_{n=0}^{N} \mathcal{H}_{\mathrm{eff}}^n \,, \tag{8}$$

i.e.

$$\mathcal{H}_{\text{eff}} v = \left( \bigoplus_{n=0}^{N} \mathcal{H}_{\text{eff}}^n \right) v = \left( \bigoplus_{n=0}^{N} \mathcal{H}_{\text{eff}}^n v_n \right), \tag{9}$$

for $v = \sum_{n=0}^{N} v_n$ and $v_n \in \mathscr{H}_{\text{eff}}^n$. The block-diagonal form of $\mathcal{H}_{\text{eff}}$ is specified by demanding that $\mathcal{H}_{\text{eff}}^n$ contains the eigenstates adiabatically connected to the eigenstates of $\mathcal{H}_0^n$. The set of (possibly degenerate) energies of those eigenstates is denoted by $e^n$.

After having defined the block-diagonalised form of the effective Hamiltonian (8) resulting from a unitary transformation $T$, we next introduce the concept of cluster-additivity for such transformations. Historically, first linked-cluster expansions for perturbative ground-state energy calculations were performed in 1955 [26] and applied to calculate zero-temperature ground state properties in high orders later in the 1980s using Nickel's cluster expansion method from unpublished work [21, 27]. The transformation used to calculate ground-state properties is not important since the ground-state additivity

$$e^0(A \cup B) = e^0(A) + e^0(B) \tag{10}$$

is always fulfilled for disconnected clusters $A$ and $B$ assuming a non-degenerate ground-state subspace. With Nickel's cluster expansion method, even excitation gaps could be calculated [4] by grouping terms in orders of the number of sites of the lattice, although a restriction to linked clusters was not sufficient for that. Still, these calculations were more efficient than calculations on linked clusters using a cluster-additive transformation [28] due to the higher efficiency of the method. The proper formalism to derive the right cluster-additive part of the effective one-particle Hamiltonian was written down in 1996 by Gelfand [29]. A more extensive review can be found in [30]. The decisive point was to not do a linked-cluster expansion for the effective Hamiltonian in the one-particle space $\mathcal{H}_{\text{eff}}^1$ but to the effective Hamiltonian minus the ground-state energy:

$$\bar{\mathcal{H}}_{\text{eff}}^1(A \cup B) \equiv \mathcal{H}_{\text{eff}}^1(A \cup B) - e^0(A \cup B) = \bar{\mathcal{H}}_{\text{eff}}^1(A) \oplus \bar{\mathcal{H}}_{\text{eff}}^1(B). \tag{11}$$

In contrast to $\mathcal{H}_{\text{eff}}^1$, $\bar{\mathcal{H}}_{\text{eff}}^1$ is additive. This was generalized to a proper cluster expansion for two particles around 2000 [2, 7, 31] and was further generalized to multi-particle excitations in 2003 [32]. They introduced the notion of cluster additivity: An effective cluster additive Hamiltonian takes the form

$$\mathcal{H}_{\text{eff}}(A \cup B) = \mathcal{H}_{\text{eff}}(A) \otimes \mathbb{1}_B + \mathbb{1}_A \otimes \mathcal{H}_{\text{eff}}(B), \tag{12}$$

on disconnected parts $A$ and $B$ of the lattice. We stress that this form is different to the direct sum in Eq. (3). However, if the effective Hamiltonian takes the cluster-additive form of Eq. (12), it can be decomposed into additive parts and a linked-cluster expansion can be performed. These additive parts, denoted by $\bar{\mathcal{H}}_{\text{eff}}^n$, are inductively defined by

$$
\begin{aligned}
\mathcal{H}_{\text{eff}}^0 &= \bar{\mathcal{H}}_{\text{eff}}^0, \\
\mathcal{H}_{\text{eff}}^1 &= \bar{\mathcal{H}}_{\text{eff}}^0|_1 + \bar{\mathcal{H}}_{\text{eff}}^1|_1, \\
&\;\;\vdots \\
\mathcal{H}_{\text{eff}}^N &= \sum_{n=0}^{N} \bar{\mathcal{H}}_{\text{eff}}^n|_N.
\end{aligned}
\tag{13}
$$

The first two equations are precisely what was described by Gelfand [29]. To understand the action of $\bar{\mathcal{H}}_{\text{eff}}^m|_n$ on a state one has to expand the state in the position basis. Then, for each position basis state, one finds all product state decompositions into two position basis states. $\bar{\mathcal{H}}_{\text{eff}}^m|_n$ then acts with an identity on the one part of the product state having unperturbed energy $e_0^n - e_0^m$ in $\mathcal{H}_0$, and with $\bar{\mathcal{H}}_{\text{eff}}^m|_m$ on the other part.

## 2.2 Cluster-additive block diagonalisation methods

The subtractions of Eq. (13) are necessary to perform linked-cluster expansions but not sufficient. For degenerate subspaces of $\mathcal{H}_0$, the transformation used is not uniquely determined and the cluster-additivity property of (12) is not necessarily given. There are two prominent approaches to construct cluster-additive effective Hamiltonians. Both make use of the linking structure of the commutator.

The first one is the method of continuous unitary transformations (CUTs), which are defined by the flow equations

$$\partial_l \mathcal{H} = [\eta, \mathcal{H}], \tag{14}$$

with $\eta(l)$ the anti-Hermitian generator of the transformation. In physics they were introduced 1993 by Wegner [33] and Glazek and Wilson [34] with the double-bracket flow, which was known in mathematics already in 1988 [35]. To use flow equations to study eigenvalue problems was already proposed by Rutishauser in 1954 with an infinitesimal version of the QR algorithm [36]. The Toda flow is another famous flow known from the study of the Toda lattice in statistical mechanics [37]. Its relation to a matrix flow for tridiagonal matrices was understood by Flachka and Moser in 1974 and 1975 [38,39]. This flow was generalized and applied to banded matrices by Mielke 1998 [40]. Stein was one of the first to solve continuous unitary transformations of that flow perturbatively in 1997 [41] and the flow was generalized further by Knetter and Uhrig in 2000, where they introduced the quasi-particle generator $\eta_{\mathrm{QP}}$ [1]. They obtained a general perturbative solution for this flow equation under the special condition of an equidistant spectrum of $\mathcal{H}_0$ and called it perturbative continuous unitary transformations (pCUT). In an eigenbasis of $\mathcal{H}_0$ the quasi-particle generator $\eta_{\mathrm{QP}}$ can be defined as

$$\eta_{\mathrm{QP},i,j}(l) = \mathrm{sgn}(\mathcal{H}_{0,i,i} - \mathcal{H}_{0,j,j})\mathcal{H}_{i,j}(l). \tag{15}$$

By stating $\mathcal{H}(0)$ is linked we define what processes are considered as linked. The off-diagonal parts of $\mathcal{H}(0)$ are assumed to be local operators. Two local operators commute when they act on disconnected parts of the lattice. As $\eta_{\mathrm{QP}}(0)$ decouples all blocks of $\mathcal{H}(0)$, it is also linked and can be written as a sum of local operators. Then by definition of the flow equation (14), the cluster-additivity property is ensured during the flow as the commutator vanishes for local operators acting on disconnected clusters.

The second genuinely linked-cluster transformation is the multi-block orthogonalization method (MBOT) [2,7]. A similar construction can also be found in [42]. As the name indicates, also here it is crucial that all blocks of the Hamiltonian are decoupled. This transformation is constructed with the matrix exponential and a global generator $\mathcal{S}$, i.e. $T = \exp(-\mathcal{S})$. It makes use of the connection between Lie algebra and matrix exponential as well as the linked structure established by the commutator expansion

$$\exp(\mathcal{S})\mathcal{H}\exp(-\mathcal{S}) = \sum_{n=0}^{\infty} \frac{[(\mathcal{S})^n, \mathcal{H}]}{n!}, \quad \text{where} \quad [(\mathcal{S})^n, \mathcal{H}] \equiv [\underbrace{\mathcal{S}, \dots [\mathcal{S}, [\mathcal{S}, \mathcal{H}]] \dots}_{n \text{ times}}]. \tag{16}$$

It is constructed order by order demanding that up to a given order all off-diagonal elements between different blocks of $\mathcal{H}_{\mathrm{eff}}$ vanish. As the first-order part of $\mathcal{S}$ has to decouple all blocks, it can be written as a sum of local operators. From the form of (16), it is then ensured that the transformation is linked cluster in the next order if $\mathcal{S}$ contains only linked terms in all previous orders. For the sake of completeness, we mention that in [42] also a local transformation constructed order by order as

$$T = \exp(-\lambda \mathcal{S}_1) \cdot \dots \cdot \exp(-\lambda^n \mathcal{S}_n) \tag{17}$$

is introduced.

Both pCUT and MBOT can be constructed order by order in a model-independent form for Hamiltonians with equidistant $\mathcal{H}_0$. There is also a model-dependent method to use $\eta_{QP}$ perturbatively (epCUT) and non-perturbatively (deepCUT) [43] for $\mathcal{H}_0$ with a non-equidistant spectrum directly in the thermodynamic limit. Also, recently an extension of the pCUT approach to multiple quasiparticle types as well as non-Hermitian Hamiltonians and open systems was introduced under the name pcst$^{++}$ [44]. It should also be possible to write down model-independent perturbative expressions for MBOT and $\mathcal{H}_0$ with non-equidistant spectrum similarly as in the Schrieffer-Wolff expansion of the minimal transformation but now using projectors on all eigenspaces of $\mathcal{H}_0$. Unfortunately, it is hard to transfer the MBOT method to non-perturbative exact calculations on finite graphs since it is difficult to find a transformation that sets all block-diagonal parts of $\mathcal{S}$ to zero while block-diagonalising the Hamiltonian. Also how to efficiently truncate the basis states for MBOT is not clear non-perturbatively. In contrast, the application of flow equations using $\eta_{QP}$ to non-perturbative problems on finite systems is straightforward and was used in the gCUT approach [24]. With regard to basis truncations it is important to realize that one can use a modified version of the generator $\eta_{QP}$

$$\eta^n_{QP,i,j}(l) = \left(1 - \Theta(\mathcal{H}_{0,i,i} - e_0^{n+1})\Theta(\mathcal{H}_{0,j,j} - e_0^{n+1})\right)\text{sgn}(\mathcal{H}_{0,i,i} - \mathcal{H}_{0,j,j})\mathcal{H}_{i,j}(l), \qquad (18)$$

and still obtain the same effective Hamiltonian in the blocks $m \leq n$ [45]. To see this we introduce the set of indices in the $n$-particle block $s_n$. Then we note that the special form of $\eta_{QP}$ leaves the flow in lower subspaces $m \leq n$ invariant under unitary transformations of the higher subspaces $m > n$ as can be seen by

$$\sum_k \mathcal{H}_{i,k}(l)\mathcal{H}_{k,j}(l) = \sum_k (\mathcal{H}U_{i,k})(l)(U^\dagger\mathcal{H}(l))_{k,j}, \qquad (19)$$

with $i, j$ in the subspaces $\bigcup_{m \leq n} s_m$ and $k$ in the higher-energy spaces $\bigcup_{m > n} s_m$ and $U$ a unitary matrix acting on the states $k$. As a consequence, one can efficiently truncate the basis states using the Krylov subspace of $\bigoplus_{m=0}^n \mathcal{H}_0^m$ when targeting the subspace $n$ of $\mathcal{H}_{eff}$ with the quasiparticle generator because states of higher orders of the Krylov subspace only contribute at larger times $l$ of the flow. This efficient way of truncating is a big advantage of the special form of $\eta_{QP}$ and distinguishes this generator. With this, we conclude the discussion of existing cluster-additive block-diagonalisation methods.

## 2.3  Projective block-diagonalisation method

Another type of transformation is the projective transformation $T$ constructed of the eigenstates and energies of the block $n$ of interest. This transformation can be given in an order-independent form, needs minimal information to be constructed, has minimal norm $\|\mathbb{1} - T\|$ and in many situations can be implemented numerically more efficiently than the transformations discussed in the last subsection because only matrix-vector multiplications are needed and for most cases obtaining energies and eigenstates with Krylov-based algorithms is faster than solving differential equations. Unfortunately, it only allows for a linked-cluster expansion of excitations under special circumstances.

The projective transformation is constructed by projectors $P_n$ on the eigenspaces of $\mathcal{H}_0$ and projectors $\bar{P}_n$ on the adiabatically connected eigenspaces of $\mathcal{H}_{eff}$. Projectors are idempotent operators, i.e. $P_n^2 = P_n$ and $\bar{P}_n^2 = \bar{P}_n$. For $v \in \mathcal{H}$

$$P_n v \in \mathcal{H}_0^n, \qquad (20)$$

and

$$\bar{P}_n v \in \mathcal{H}_{eff}^n. \qquad (21)$$

Further, from the orthogonality of the subspaces the resolution of identity ,

$$\mathbb{1} = \sum_n P_n = \sum_n \bar{P}_n \,, \tag{22}$$

follows. A good educational introduction to perturbation theory described in the framework of projection operators is given in [46].

We first state the form of the projective transformation introduced by Takahashi [18]:

$$T = \sum_n T_n \,, \tag{23}$$

$$T_n = \bar{P}_n P_n \left( \sum_m P_m \bar{P}_m P_m \right)^{-1/2} \,, \tag{24}$$

He further used a result of Kato [47] for the perturbative form of the projector $\bar{P}_n$

$$\bar{P}_n = P_n - \sum_{s=1}^{\infty} \sum_{k_1+\cdots+k_{s+1}=s, \, k_i \leq 0} S_n^{k_1} V S_n^{k_2} V \dots V S_n^{k_{s+1}} \,, \tag{25}$$

where $S_n^0 \equiv -P_n$, $S_n^k \equiv \left( \frac{1-P_n}{e_0^n - \mathcal{H}_0} \right)^k$ and realized that $P_n \left( \sum_m P_m \bar{P}_m P_m \right)^{-1/2} P_n$ can be expanded similarly using Kato's expression. Note that while $P_n \bar{P}_n P_n$ can not be inverted its restriction to the subspace $\mathcal{H}_0^n$ can. The local expressibility of the transformation is important as it shows that the transformation has no contributions on subgraphs of the lattice with a larger number of bonds than the perturbation order. The transformation $T$ is symmetric in the diagonal blocks as can be seen by

$$P_n T P_n = P_n T_n P_n = P_n \bar{P}_n P_n \left( \sum_m P_m \bar{P}_m P_m \right)^{-1/2} P_n = P_n \left( \sum_m P_m \bar{P}_m P_m \right)^{1/2} P_n \,, \tag{26}$$

and

$$P_n T^\dagger P_n = P_n T_n^\dagger P_n = P_n \left( \sum_m P_m \bar{P}_m P_m \right)^{-1/2} P_n \bar{P}_n P_n = P_n \left( \sum_m P_m \bar{P}_m P_m \right)^{1/2} P_n \,. \tag{27}$$

This shows the equivalence of the perturbative expansion of $T$ with the two-block orthogonalization method (TBOT) [3] as for TBOT in [3] it was shown that any perturbative transformation that decouples two blocks of the Hamiltonian is uniquely determined by demanding symmetric diagonal blocks.

The projective transformation can also be written in the form of a Schrieffer-Wolff transformation $T_{\text{SW}} = \exp(-\mathcal{S}_{\text{SW}})$ that decouples block $n$ from the rest. We understand as a Schrieffer-Wolff transformation $T_{\text{SW}}$ any transformation with a particular anti-block-diagonal form of $\mathcal{S}_{\text{SW}}$. Introducing

$$R = \sum_{m, \, m \neq n} P_m \,, \tag{28}$$

it can be written as

$$T_{\text{SW}} = \left( \bar{P}_n P_n + \bar{R} R \right) \left( P_n \bar{P}_n P_n + R \bar{R} R \right)^{-1/2} = \exp(-\mathcal{S}_{\text{SW}}) \,, \tag{29}$$

where $\mathcal{S}_{\text{SW}}$ takes the form

$$\mathcal{S}_{\text{SW}} = \begin{pmatrix} 0 & \mathcal{S}_{\text{SW},n,R} \\ -\mathcal{S}_{\text{SW},n,R}^\dagger & 0 \end{pmatrix} \,. \tag{30}$$

That $\mathcal{S}_{\text{SW}}$ has to take such a form follows at least perturbatively from the uniqueness of $\mathcal{S}_{\text{SW}}$, the symmetry of $T_{\text{SW}}$ in its diagonal blocks, and the fact that an exponential of an anti-block diagonal $\mathcal{S}_{\text{SW}}$ as in Eq. (30) yields a transformation that is symmetric in the diagonal blocks. In [19] the transformation is constructed perturbatively by an $\mathcal{S}_{\text{SW}}$ of that form and it is called canonical form of van Vleck perturbation theory. A review of the Schrieffer-Wolff transformation also constructs the transformation order by order this way [48], while also giving a very convenient form of the transformation as direct rotation

$$T_{\text{SW}} = \sqrt{(\bar{P}_n - \bar{R})(P_n - R)}, \tag{31}$$

between $P_n$ and $\bar{P}_n$, i.e.

$$T_{\text{SW}}^{\dagger} \bar{P}_n T_{\text{SW}} = P_n. \tag{32}$$

The equivalence between (29) and (31) is most easily seen by comparing

$$\left(\bar{P}_n P_n + \bar{R}R\right)^2 = \bar{P}_n P_n \bar{P}_n P_n + \bar{R}R\bar{R}R + \bar{P}_n P_n \bar{R}R + \bar{R}R\bar{P}_n P_n, \tag{33}$$

and

$$(\bar{P}_n - \bar{R})(P_n - R)\left(P_n \bar{P}_n P_n + R\bar{R}R\right) = \bar{P}_n P_n \bar{P}_n P_n + \bar{R}R\bar{R}R - \bar{P}_n R\bar{R}R - \bar{R}P_n \bar{P}_n P_n. \tag{34}$$

The expressions are identical since $\mathbb{1} = P_n + R$ and $\bar{P}_n \bar{R} = 0$. In [48] the transformation is constructed perturbatively order by order using the form of the matrix exponential Eq. (30). This is not necessary as Takahashi's form of the transformation for the effective low-energy block is exactly identical and can be written down non-inductively. Another unique property of $T_{\text{SW}}$ is that it has minimal norm $\|\mathbb{1} - T_{\text{SW}}\|$ of all possible transformations that decouple the block $n$ from the rest [48, 49]. In contrast to the MBOT transformation, the global generator only is anti-block-diagonal with respect to two blocks and because of that has non-local anti-block-diagonal terms in general.

At last we state the form of the transformation given in [20]. It is very similar to Takahashi's form but given in terms of eigenvectors instead of projectors. This form will be particularly useful for the construction of the cluster-additive projective transformation in Sec. 3. The eigenvectors and energies $X_0$ and $D_0$ of $\mathcal{H}_0$ and $X$ and $D$ of $\mathcal{H}$ fulfil

$$\mathcal{H}X_0 = X_0 D_0, \tag{35}$$

and

$$\mathcal{H}X = XD. \tag{36}$$

Projection operators and eigenvectors are related by

$$P_{n,i,j} = \sum_{k \in s_n} X_{0,i,k} X_{0,k,j}^{\dagger}, \tag{37}$$

and

$$\bar{P}_{n,i,j} = \sum_{k \in s_n} X_{i,k} X_{k,j}^{\dagger}, \tag{38}$$

where the ordering of basis states and energies is such that $X_{0,i,j}$ is only non-zero for $i, j \in s_n$. Here we remind that the set of indices in the $n$-particle block is denoted by $s_n$. Introducing

$$X^{P_n} \equiv P_n X P_n, \tag{39}$$

one can then write the transformation as

$$T_{n,i,j} = \sum_k X_{i,k} \left( X^{P_n \dagger} \left( \sum_m X^{P_m} X^{P_m \dagger} \right)^{-1/2} \right)_{k,j}, \tag{40}$$

with $k \in s_n$. In [20] it was proved that this transformation has minimal norm $\|\mathbb{1} - T\|$, which shows that also when one wants to decouple all blocks and not just two as in $T_{\text{SW}}$ this is the transformation with minimal norm. The MBOT method, which is a Schrieffer-Wolff transformation of local anti-block-diagonal operators, is different and consequently does not have minimal norm. Hence, only when one decouples two blocks an anti-block-diagonal $\mathcal{S}_{\text{SW}}$ leads to a transformation with minimal norm $\|\mathbb{1} - T_{\text{SW}}\|$.

For the effective Hamiltonian in the desired block $n$ only the part $X^{P_n} X^{P_n \dagger}$ contributes. By denoting the restriction of $X^{P_n}$ to the basis states $s_n$ with $X^{P_n}_{s_n}$ the part of the transformation that creates the effective Hamiltonian in block $n$ can be written as

$$T_{n,i,s_n} = \sum_{k \in s_n} X_{i,k} \left( X^{P_n \dagger}_{s_n} \left( X^{P_n}_{s_n} X^{P_n \dagger}_{s_n} \right)^{-1/2} \right)_{k,s_n}. \tag{41}$$

As these are the only basis states for which $X^{P_n}$ has non-zero matrix elements this restricts the transformation to the relevant part for each block and can help make considerations easier. In particular, for two disconnected clusters $A$ and $B$ and transformations $T_{l,A}$ in $A$ and $T_{k,B}$ in $B$ and a transformation $T_{n,s_l \otimes s_k}$ on $A \cup B$ in the subspace $n$, that projects only on the states $s_l \otimes s_k$ (but only on these, not on the whole block $n$ on $A \cup B$), one finds

$$H_{\text{eff},s_l \otimes s_k}(A \cup B) = H_{\text{eff},s_l}(A) \otimes \mathbb{1}_B + \mathbb{1}_A \otimes H_{\text{eff},s_k}(B), \tag{42}$$

as

$$\sum_{i,j} X^{\dagger}_{s_l \otimes s_k, i} \mathcal{H}_{i,j} X_{j,s_l \otimes s_k} = D_{s_l}(A) \otimes \mathbb{1}_B + \mathbb{1}_A \otimes D_{s_k}(B), \tag{43}$$

and

$$\left( X^{P_n \dagger}_{s_l \otimes s_k} \left( X^{P_n}_{s_l \otimes s_k} X^{P_n \dagger}_{s_l \otimes s_k} \right)^{-1/2} \right) = \left( X^{P_l \dagger}_{s_l} \left( X^{P_l}_{s_l} X^{P_l \dagger}_{s_l} \right)^{-1/2} \right) \otimes \left( X^{P_k \dagger}_{s_k} \left( X^{P_k}_{s_k} X^{P_k \dagger}_{s_k} \right)^{-1/2} \right), \tag{44}$$

where $e^n_0 - e^0_0 = (e^l_0 - e^0_0) + (e^k_0 - e^0_0)$. This was also shown in [48] and shows that the effective Hamiltonian of the projective transformation allows performing a linked-cluster decomposition for degenerate ground states. For excitations, it is not helpful since one can not separate excitations in $A \cup B$ with one excitation in $A$ and ground state in $B$ from ground state in $A$ and one excitation in $B$. The problems caused by this will become obvious in the next subsection, where we show the failure of a linked-cluster expansion for spin-flip excitations in a simple toy model.

## 2.4 Failure of linked-cluster expansion for excited states with projective method

Gelfand realized that a linked-cluster expansion for elementary excitations is possible with non-cluster additive transformations as long as the elementary excitations have a different quantum number than the ground state [29]. To show the failure of a linked-cluster expansion for the minimal transformation we therefore consider a high-field expansion of the Hamiltonian given as the sum of the transverse-field Ising chain, where this is given, and a parity breaking term $\sigma^z_\nu \sigma^x_{\nu+1}$:

$$\mathcal{H} = \sum_\nu \sigma^z_\nu + \sum_\nu \left( \lambda \sigma^x_\nu \sigma^x_{\nu+1} + \mu \left( \sigma^z_\nu \sigma^x_{\nu+1} + \sigma^x_\nu \sigma^z_{\nu+1} \right) \right). \tag{45}$$

The Pauli matrices $\sigma^{x/z}_\nu$ describe spins-1/2 on site $\nu$. For $\mu \neq 0$ ground state and spin-flip excitations are coupled to each other. Now we consider two disconnected clusters $A$ and $B$. The Hamiltonian on $A \cup B$ can be written as

$$\mathcal{H} = \mathcal{H}_A + \mathcal{H}_B\,, \tag{46}$$

where

$$[\mathcal{H}_A, \mathcal{H}_B] = 0 \tag{47}$$

holds. Consequently the eigenfunctions of $H_{A \cup B}$ take the form

$$|\Psi\rangle_{A \cup B} = |\Psi\rangle_A \otimes |\Psi\rangle_B\,, \tag{48}$$

and have an energy

$$\mathcal{H} |\Psi\rangle = (\mathcal{H}_A |\Psi\rangle_A) \otimes |\Psi\rangle_B + |\Psi\rangle_A \otimes (\mathcal{H}_B |\Psi\rangle_B) = (e_A + e_B) |\Psi\rangle\,. \tag{49}$$

For spin-flip excitations on $A \cup B$ it follows that they are either build of a ground state on $A$ and a spin-flip excitation on $B$ or vice versa:

$$|\Psi\rangle_{1,A \cup B} = |\Psi\rangle_{1,A} \otimes |\Psi\rangle_{0,B} \quad \vee \quad |\Psi\rangle_{1,A \cup B} = |\Psi\rangle_{0,A} \otimes |\Psi\rangle_{1,B}\,. \tag{50}$$

For the case $\mu = 0$ where the parity is not broken, $P_0 |\Psi\rangle_1 = 0$. Then $X_{s_1}^{P_1}$ is block-diagonal in the $A$- and $B$-blocks

$$X_{s_1}^{P_1} = \begin{pmatrix} X_{s_1,A}^{P_1} X_{s_0,B}^{P_0} & 0 \\ 0 & X_{s_1,B}^{P_1} X_{s_0,A}^{P_0} \end{pmatrix}, \tag{51}$$

and additivity of $\bar{\mathcal{H}}_{\text{eff}}^1$ is given

$$T_1^\dagger \mathcal{H} T_1 - e^0(A \cup B) = \bar{\mathcal{H}}_{\text{eff}}^1(A \cup B) = \bar{\mathcal{H}}_{\text{eff}}^1(A) \oplus \bar{\mathcal{H}}_{\text{eff}}^1(B)\,. \tag{52}$$

This is not the case when $\mu \neq 0$. Then $P_0 |\Psi\rangle_1 \neq 0$ and $X_{s_1}^{P_1}$ is not block-diagonal in the $A$- and $B$-blocks any more

$$X_{s_1}^{P_1} = \begin{pmatrix} X_{s_1,A}^{P_1} X_{s_0,B}^{P_0} & X_{s_1,A}^{P_0} X_{s_0,B}^{P_1} \\ X_{s_1,B}^{P_0} X_{s_0,A}^{P_1} & X_{s_1,B}^{P_1} X_{s_0,A}^{P_0} \end{pmatrix}. \tag{53}$$

Consequently, additivity of $\bar{\mathcal{H}}_{\text{eff}}^1$

$$T_1^\dagger \mathcal{H} T_1 - e^0(A \cup B) = \bar{\mathcal{H}}_{\text{eff}}^1(A \cup B) \neq \bar{\mathcal{H}}_{\text{eff}}^1(A) \oplus \bar{\mathcal{H}}_{\text{eff}}^1(B) \tag{54}$$

is not given any more. If one performs calculations for the model with $\mu = 1$ one finds these non-linked terms in order four. Particles can then hop between disconnected clusters as illustrated in Fig. 1, which is never allowed in a linked-cluster expansion. The crucial step for the construction of a cluster additive projective transformation is to modify $X_{s_1}^{P_1}$ to restore block-diagonal form for the general case $\mu \neq 0$ and to eliminate these hopping elements between disconnected clusters.

## 3 Projective cluster-additive transformation

In the last section we reviewed the minimal projective transformation and showed an example where the failure of linked-cluster expansion for excited states was shown. In particular, the problem could be seen in the non-block diagonal form of $X_{s_1}^{P_1}$ in (53). It is the major achievement of this paper to introduce the projective cluster-additive transformation $T^{\text{pca}}$ which cures this problem.

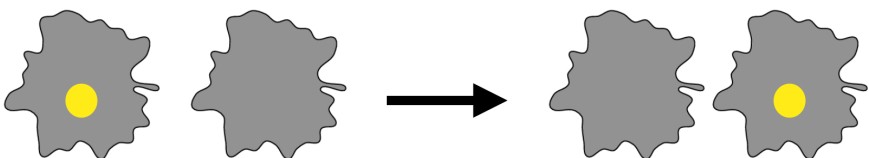

Figure 1: The figure depicts a hopping process of one particle (yellow ball) between two disconnected clusters. For the Hamiltonian (45) such hopping elements are seen in the effective one-particle Hamiltonian in order four of perturbation. These processes are a manifestation of the violation of cluster-additivity of the minimal projective transformation.

## 3.1 Cluster-additivity for single-particle states

It is necessary to modify $X_{s_1}^{P_1}$ to $\tilde{X}_{s_1}^{P_1}$ to obtain a cluster-additive transformation for single-particle states. To achieve this we modify the eigenstates of $\mathcal{H}$. For ground-state energies additivity is always given and consequently, the ground state $|\Psi\rangle_0$ is not modified:

$$|\tilde{\Psi}\rangle_0 = |\Psi\rangle_0 \,. \tag{55}$$

For single-particle eigenstates $|\Psi\rangle_1$ we modify in the following way,

$$|\tilde{\Psi}\rangle_1 = |\Psi\rangle_1 - (1/\langle 0|\Psi_0\rangle)\langle 0|\Psi_1\rangle\,|\Psi\rangle_0 \,, \tag{56}$$

where $|0\rangle$ denotes the unperturbed ground state. Note that in general the states $|\tilde{\Psi}\rangle_1$ as well as $|\tilde{\Psi}\rangle_0$ and $|\tilde{\Psi}\rangle_1$ are not orthogonal and normalized any more. The ground-state subtraction of $|\Psi\rangle_0$ in $|\tilde{\Psi}\rangle_1$ leads to

$$P_0\,|\tilde{\Psi}\rangle_1 = 0\,. \tag{57}$$

As long as $\langle 0|\Psi_0\rangle \neq 0$ this subtraction is unique. Recalling the form (50) of a single-particle eigenstate on two disconnected clusters $A \cup B$ we find

$$|\tilde{\Psi}\rangle_{1,A\cup B} = |\tilde{\Psi}\rangle_{1,A} \otimes |\tilde{\Psi}\rangle_{0,B} \,. \tag{58}$$

$\tilde{X}_{s_1}^{P_1}$ then takes the form

$$\tilde{X}_{s_1}^{P_1} = \begin{pmatrix} \tilde{X}_{s_1,A}^{P_1}\tilde{X}_{s_0,B}^{P_0} & 0 \\ 0 & \tilde{X}_{s_1,B}^{P_1}\tilde{X}_{s_0,A}^{P_0} \end{pmatrix}, \tag{59}$$

because $\tilde{X}_{s_1,A}^{P_0} = \tilde{X}_{s_1,B}^{P_0} = 0$. The linked-cluster transformation of the single-particle block can now be conveniently written as

$$T_{1,i,s_1}^{\mathrm{pca}} = \sum_{k\in s_1} X_{i,k}\left(\tilde{X}_{s_1}^{P_1\,\dagger}\left(\tilde{X}_{s_1}^{P_1}\ \tilde{X}_{s_1}^{P_1\,\dagger}\right)^{-1/2}\right)_{k,s_1}. \tag{60}$$

Particularly important is the part

$$\mathcal{T}_{1,s_1,s_1}^{\mathrm{pca}} = \left(\tilde{X}_{s_1}^{P_1\,\dagger}\left(\tilde{X}_{s_1}^{P_1}\ \tilde{X}_{s_1}^{P_1\,\dagger}\right)^{-1/2}\right)_{s_1,s_1}, \tag{61}$$

since its form determines the matrix elements of $\mathcal{H}_{\mathrm{eff}}^n$. As we have already seen, this part is block-diagonal

$$\mathcal{T}_{1,A\cup B}^{\mathrm{pca}} = \mathcal{T}_{1,A}^{\mathrm{pca}} \oplus \mathcal{T}_{1,B}^{\mathrm{pca}} \,. \tag{62}$$

The other part of the transformation just yields a diagonal matrix

$$\sum_{i,j} X^{\dagger}_{s_1,i} \mathcal{H}_{i,j} X_{j,s_1} = D_A \oplus D_B \,. \tag{63}$$

Combining the direct sum of eigenvalues on $A \cup B$

$$D_A \oplus D_B - e^0(A \cup B) = e^1_A \oplus e^1_B \,, \tag{64}$$

with the form of $\mathcal{T}^{\mathrm{pca}}_1$ in Eq. (62) one obtains additivity of $\bar{\mathcal{H}}^1_{\mathrm{eff}}$:

$$\sum_{r,k} T^{\mathrm{pca},\dagger}_{1,s_1,r} \mathcal{H}_{r,k} T^{\mathrm{pca}}_{1,k,s_1} - e^0(A \cup B) = \bar{\mathcal{H}}^1_{\mathrm{eff}}(A \cup B) = \bar{\mathcal{H}}^1_{\mathrm{eff}}(A) \oplus \bar{\mathcal{H}}^1_{\mathrm{eff}}(B) \,. \tag{65}$$

For one-particle excitations we now have constructed the right transformation. The more general case of multi-particle excitations will be discussed in the next subsection.

## 3.2 Cluster-additivity for multi-particle excitations

As mentioned before, the cluster additivity of the effective Hamiltonian implies that we can construct additive irreducible operators in every block of interest of the effective Hamiltonian. To show cluster-additivity for multi-particle excitations we again make use of the tensor product structure of eigenstates on $A \cup B$ with $A$ and $B$ not connected for $n$-particle states $|\Psi\rangle_n$ with energy $e^a_{0,A} + e^b_{0,B} = e^n_{0,A\cup B}$ of $\mathcal{H}_0$:

$$|\Psi\rangle_{n,A\cup B} = |\Psi\rangle_{a,A} \otimes |\Psi\rangle_{b,B} \,. \tag{66}$$

What changes compared to single-particle excitations is the transformation of eigenstates $|\Psi\rangle \to |\tilde{\Psi}\rangle$ for the construction of the transformation. For a state with energy $e^n_0$ we demand that the projection on eigenstates of $\mathcal{H}_0$ with $e^m_0 < e^n_0$ is zero, i.e. for

$$R = \sum_{m,\,m<n} P_m \,, \tag{67}$$

we need to have

$$R|\tilde{\Psi}\rangle_n = 0 \,. \tag{68}$$

This has to be achieved by subtracting lower-energy eigenstates of $|\tilde{\Psi}\rangle_n$. As long as

$$Y_{n-1} = X_{i,j} \,, \quad i,j \in \cup_{m<n} s_m \,, \tag{69}$$

is invertible the construction is always possible and unique. Assuming non-singular $Y_{n-1}$, the transformed states $|\tilde{\Psi}\rangle_n$ are defined as

$$|\tilde{\Psi}\rangle_n = |\Psi\rangle_n - \sum_{m<n} \left[ Y^{-1}_{n-1} (R|\Psi\rangle_n) \right]_m |\Psi\rangle_m \,. \tag{70}$$

The singular values of $Y_{n-1}$ are the square roots of the eigenvalues of

$$W_{n-1} = \sum_{m<n} P_m \sum_{m<n} \bar{P}_m \sum_{m<n} P_m \,. \tag{71}$$

As we discuss later in the context of NLCEs (see Subsec. 4.2), particle decay highly influences the convergence properties of the non-perturbative expansion. For particle-decay of $n$-particle states it is important to investigate the behaviour of $W_n$ and not of $W_{n-1}$. The reason is that

particle-decay of the $n$-particle states would show up as a problem in the construction of $m$-particle states with $m > n$. When the smallest eigenvalue of $W_n$ drops to almost zero sharply, this is a hallmark of particle-decay. The transformation from $|\Psi\rangle_n$ to $|\tilde{\Psi}\rangle_n$ can be visualized as

$$
\begin{pmatrix} P_0 |\Psi\rangle_n \\ \vdots \\ P_n |\Psi\rangle_n \\ \vdots \\ P_N |\Psi\rangle_n \end{pmatrix} \rightarrow \begin{pmatrix} 0 \\ \vdots \\ P_n |\tilde{\Psi}\rangle_n \\ \vdots \\ P_N |\tilde{\Psi}\rangle_n \end{pmatrix} . \tag{72}
$$

Since this subtraction is unique for non-singular $Y_{N-1}$ in Eq. (69), it follows

$$
|\tilde{\Psi}\rangle_{n,A\cup B} = |\tilde{\Psi}\rangle_{a,A} \otimes |\tilde{\Psi}\rangle_{b,B} . \tag{73}
$$

Eq. (73) is at the heart of the cluster-additivity of the transformation. It follows

$$
\tilde{X}^{P_n}_{s_a \otimes s_b} = \tilde{X}^{P_a}_{s_a,A} \otimes \tilde{X}^{P_b}_{s_b,B} , \tag{74}
$$

and with that for the transformation

$$
\tilde{X}^{P_n \dagger}_{s_a \otimes s_b} \left( \tilde{X}^{P_n}_{s_a \otimes s_b} \tilde{X}^{P_n \dagger}_{s_a \otimes s_b} \right)^{-1/2} = \tilde{X}^{P_a \dagger}_{s_a,A} \left( \tilde{X}^{P_a}_{s_a,A} \tilde{X}^{P_a \dagger}_{s_a,A} \right)^{-1/2} \otimes \tilde{X}^{P_b \dagger}_{s_b,B} \left( \tilde{X}^{P_b}_{s_b,B} \tilde{X}^{P_b \dagger}_{s_b,B} \right)^{-1/2} . \tag{75}
$$

Then with

$$
\sum_{i,j} X^{\dagger}_{s_a \otimes s_b,i} \mathcal{H}_{i,j} X_{j,s_a \otimes s_b} = D_{s_a,A} \otimes 1_B + 1_A \otimes D_{s_b,B} , \tag{76}
$$

cluster-additivity of the transformation is a consequence of

$$
\mathcal{A}^{\dagger} \left( D_{s_a,A} \otimes 1_B + 1_A \otimes D_{s_b,B} \right) \mathcal{A} = \mathcal{H}^a_{\text{eff}}(A) \otimes 1_B + 1_A \otimes \mathcal{H}^b_{\text{eff}}(B), \tag{77}
$$

where $\mathcal{A} = \left( \tilde{X}^{P_n \dagger}_{s_a \otimes s_b} \left( \tilde{X}^{P_n}_{s_a \otimes s_b} \tilde{X}^{P_n \dagger}_{s_a \otimes s_b} \right)^{-1/2} \right)$. The transformation as a whole acting on all particle blocks can also be written down and is given as

$$
T^{\text{pca}} = X \left( \sum_m \tilde{X}^{P_m} \right)^{\dagger} \left( \left( \sum_m \tilde{X}^{P_m} \right) \left( \sum_m \tilde{X}^{P_m} \right)^{\dagger} \right)^{-1/2} . \tag{78}
$$

with $\tilde{X}^{P_n} = P_n \tilde{X} P_n$.

## 3.3 Explicit form of transformation in terms of projection operators

It is important to have the transformation also explicitly given in terms of projection operators as this allows for a local expression of the transformation using Kato's formula Eq.(25) and implies that reduced graph contributions are zero for graphs with more bonds than the perturbation order. For the explicit form, we first define

$$
\bar{\mathfrak{R}}_n \equiv \left( \sum_m R_m \bar{R}_m R_m \right)^{-1} \bar{R}_n , \tag{79}
$$

with

$$
R_n \equiv \sum_{m<n} P_m . \tag{80}
$$

The transformation then takes the form

$$T^{\text{pca}} = \left( \sum_m (\bar{P}_m - \bar{P}_m \bar{\mathfrak{R}}_m) P_m \right) \left( \sum_m P_m \left( (\bar{P}_m - \bar{P}_m \bar{\mathfrak{R}}_m)^\dagger (\bar{P}_m - \bar{P}_m \bar{\mathfrak{R}}_m) \right) P_m \right)^{-1/2}. \tag{81}$$

To prove the equivalence of (78) and (81) we need to find a way to express $X P_n (\tilde{X}^\dagger - X^\dagger)$ in terms of projection operators. We first note that the conditions

$$P_n (\tilde{X}^\dagger - X^\dagger) R_n = -P_n X^\dagger R_n$$

(subtractions of lower-energy states yield $R_n \tilde{X}^{P_n} = 0$) and

$$P_n (\tilde{X}^\dagger - X^\dagger) \bar{R}_n = P_n (\tilde{X}^\dagger - X^\dagger)$$

(only states with lower energy than in block $n$ are subtracted) determine $P_n(\tilde{X}^\dagger - X^\dagger)$ uniquely. We need to show that both these conditions are also fulfilled for $-P_n X^\dagger \bar{\mathfrak{R}}_n$ to show that $-\bar{P}_n \bar{\mathfrak{R}}_n = X P_n (\tilde{X}^\dagger - X^\dagger)$. The latter condition is obviously fulfilled by the construction of Eq. (79). For the first condition we note that

$$P_n X^\dagger \bar{\mathfrak{R}}_n R_n = P_n X^\dagger \left( \sum_m R_m \bar{R}_m R_m \right)^{-1} R_n \bar{R}_n R_n = P_n X^\dagger R_n. \tag{82}$$

This proves the equivalence of Eq. (78) and Eq. (81) and establishes the form of the transformation in terms of projection operators only. It is important to have shown this equivalence since perturbatively it follows that one can expand the transformation in local terms using Kato's formula.

# 4 Low-field expansion for the transverse-field Ising model on the square lattice

As an application we investigate the ferromagnetic transverse-field Ising model on the square lattice in the low-field ordered phase. The Hamiltonian of this paradigmatic model can be written down with Pauli matrices and takes the form

$$\mathcal{H} = -\frac{1}{4} \sum_{\langle \nu, \nu' \rangle} \sigma_\nu^z \sigma_{\nu'}^z + h \sum_\nu \sigma_\nu^x = \mathcal{H}_0 + hV, \tag{83}$$

with

$$\mathcal{H}_0 = -\frac{1}{4} \sum_{\langle \nu, \nu' \rangle} \sigma_\nu^z \sigma_{\nu'}^z, \tag{84}$$

and

$$V = \sum_\nu \sigma_\nu^x. \tag{85}$$

The Hamiltonian commutes with the spin-flip transformation $\prod_\nu \sigma_\nu^x$. In the ordered phase this $\mathbb{Z}_2$ symmetry is broken and the model undergoes a second-order phase transition in the $3d$ Ising universality class towards the disordered high-field phase when $h$ is increased. Good estimates of the critical point were obtained using high-field series expansions and quantum Monte Carlo simulations and yielded $h_c \approx 0.7610$ [4,50]. Best estimates of the critical exponent can be obtained using the conformal bootstrap method and quantum Monte Carlo simulations [51,52]. The first two digits of the correlation length exponent are given as $\nu = 0.63$. On finite systems the parity symmetry is not broken. To perform linked-cluster expansions one

therefore goes into a dual picture that is isospectral to the original one in the infinite system but has a unique polarized ground state for $h = 0$. As in [28] we define new pseudo-spin-1/2 degrees of freedom and new Pauli matrices

$$\tilde{\sigma}^z_\beta = \tilde{\sigma}^z_{\langle v, v' \rangle} = \sigma^z_v \sigma^z_{v'}, \tag{86}$$

that takes the eigenvalues $\pm 1$ of the Ising interaction on every bond $\langle v, v' \rangle$. This means that the degrees of freedom are located on the bonds and not on the sites any more. The dual Hamiltonian in this basis can be decomposed into an unperturbed and perturbed part in the following way:

$$\tilde{\mathcal{H}} = \tilde{\mathcal{H}}_0 + h\tilde{V}, \tag{87}$$

with

$$\tilde{\mathcal{H}}_0 = -\frac{1}{4} \sum_\beta \tilde{\sigma}^z_\beta, \tag{88}$$

and

$$\tilde{V} = \sum_s \tilde{A}_s, \tag{89}$$

where the plaquette operator $\tilde{A}$ takes the form

$$\tilde{A}_s = \prod_{\beta \in s(v)} \tilde{\sigma}^x_\beta. \tag{90}$$

The index $\beta$ runs over the four bonds $s(v)$ that are connected to the site $v$ in the original degrees of freedom.

In this section we are going to employ our transformation $T^{\text{pca}}$ to the low-field phase of the model and derive series and NLCE results for the spin-flip and bound-state excitation gap in this model. Bound states arise in this model because flipping two adjacent spins in the ground state yields a state with lower energy in $\mathcal{H}_0$ than flipping two spins further apart. We analyse the series results in the next subsection 4.1 and further calculate the same quantities non-perturbatively in subsection 4.2.

## 4.1 Perturbative results for single spin flip and bound states

Perturbative low-field expansions were most efficiently performed with a transformation of the same complexity as the minimal transformation [5]. Even though this calculation was done on a large number of also non-linked graphs - since it did not allow for a linked-cluster expansion of excitations because of couplings between ground state and excitations - it reached much higher orders than a calculation on only linked clusters with the pCUT method [28]. Our approach is thus ideal having the same complexity as the minimal transformation but allowing for a linked-cluster expansion.

We calculated graph embeddings on the square lattice using a hypergraph expansion [53] and obtained the embedding factors for all graphs with up to 13 sites in the original lattice. The elementary excitation in the low-field phase is a spin-flip. Next higher excitations are bound states adiabatically connected to two spin flips on adjacent spins. We calculated the spin-flip gap up to order 24 extending the results of [5] by 4 orders and the bound-state gap up to order 22 extending the results of [28] by 10 orders. It is possible to reach such high orders with graphs of only up to 13 sites since in the low-field expansion of excitations with $a$ spin-flips on a graph with $N$ sites the minimal order for a reduced graph contribution is $2(N-a)$. This property is also called strong-double-touch. We checked that both series agree with the known results of [5, 28].

As for our method it is only important to obtain the eigenspaces and energies of the excitation of interest and those of all excitations with lower energy, we used one of the most efficient methods for calculating eigenspaces and energies perturbatively, which is the two-block orthogonalization method (TBOT) form of the minimal transformation. A description of TBOT is given in [3]. With the information obtained this way we then construct the cluster-additive projective transformation to perform the linked-cluster expansion for both the spin-flip and bound-state gap. Almost all resources are needed for the TBOT calculation. Hence, we are as efficient as TBOT but only need to consider linked clusters making the method very efficient. We denote the series for the zero momentum single spin-flip gap by $\Delta$ and the one for the zero momentum bound-state gap by $\Delta_{\mathrm{bs}}$. They read respectively

$$
\begin{aligned}
\Delta = 2 - 3\,h^2 &+ 3.5833\,h^4 - 23.140\,h^6 + 133.22\,h^8 - 849.05\,h^{10} + 5738.0\,h^{12} \\
&- 40573\,h^{14} + 29615 \cdot 10\,h^{16} - 22157 \cdot 10^2\,h^{18} + 16906 \cdot 10^3\,h^{20} \\
&- 13105 \cdot 10^4\,h^{22} + 10292 \cdot 10^5\,h^{24}\,,
\end{aligned}
\tag{91}
$$

and

$$
\begin{aligned}
\Delta_{\mathrm{bs}} = 3 - 22.916\,h^4 &- 13.334\,h^6 + 263.64\,h^8 + 5213.1\,h^{10} - 7214.0\,h^{12} - 31023 \cdot 10\,h^{14} \\
&- 24296 \cdot 10^2\,h^{16} + 19814 \cdot 10^3\,h^{18} + 30204 \cdot 10^4\,h^{20} + 57170 \cdot 10^4\,h^{22}\,.
\end{aligned}
\tag{92}
$$

Note that we displayed the first five digits of the coefficients and did not round to the last digit. This accuracy can be guaranteed, while for more digits calculations would have needed to be performed with higher accuracy than double precision.

To analyse the behaviour of these series we used Padé and DLog-Padé extrapolations. A good and extensive review of extrapolation techniques in general and especially these two is [54]. Padé approximations are a well-established tool to enhance the convergence of a perturbative series and DLog-Padé extrapolations in particular mimic the algebraic behaviour of critical quantities in the vicinity of a quantum phase transition.

The series $\Delta$ of the gap is consistently alternating up to high orders. Many DLog-Padé extrapolations of $\Delta$ break down because of spurious poles. To estimate the reliability of DLog-Padé extrapolations it is helpful to study the convergence behaviour of the DLog-Padé families of order $[n, n + d]$ with $d$ fixed. As the series only contains even orders we made the analysis for the series in the variable $h^2$. Note that the maximum order of the series in this variable is 12. We found that only the families with $d = \pm 1$ show converging behaviour and that the family $d = 1$ appears to be better converged. For the $d = -1$ family the extrapolation of the highest order, i.e. the $[6, 5]$ DLog-Padé extrapolant, yields a critical point $h_c = 0.727$ and a critical exponent $\nu = 0.417$. From the highest-order $[5, 6]$ DLog-Padé extrapolant of the better-converged $d = 1$ family one obtains a critical point $h_c = 0.762$ and a critical exponent $\nu = 0.649$.

An extrapolation analysis of $\Delta_{\mathrm{bs}}$ is in principle also reasonable as the bound-state mode is stable and expected to close with the same critical exponent as the spin-flip gap, i.e. $\nu(\Delta) = \nu(\Delta_{\mathrm{bs}})$. Indeed, there are field theoretic calculations of Caselle et al. [55,56] predicting $\Delta_{\mathrm{bs}}/\Delta|_{h=h_c} \approx 1.8$. This quantity was also calculated with exact diagonalisation yielding a value of 1.84(3) [57]. Unfortunately, the series of the bound state $\Delta_{\mathrm{bs}}$ shows a complicated behaviour and no convergence of Padé or DLog-Padé extrapolations was found. In [28] $\Delta_{\mathrm{bs}}/\Delta$ was investigated with Padé and DLog-Padé extrapolations but only one extrapolation, the DLog-Padé $[4, 6]$, showed non-spurious behaviour and a value close to the numerical value of 1.84(3) as in [57]. Having calculated ten orders of perturbation more than in [28] one could hope that we find more extrapolations consistent with the predictions and calculations

of [55–57]. However, this is not the case and the additional orders rather show that the DLog-Padé family of the DLog-Padé [4, 6] extrapolant does not seem to converge with higher orders. At least up to the calculated orders so far, no behaviour of the series extrapolations that is consistent with the expectation of $\Delta_{\mathrm{bs}}/\Delta|_{h=h_c} \approx 1.8$ could be found.

## 4.2 Non-perturbative results for single spin flip and bound states

Non-perturbative linked-cluster expansions (NLCEs) for the low-field phase of the transverse-field Ising model were so far only performed for ground-state energies and ground-state expectation values of observables [58, 59]. In these papers the linked-cluster expansion for the ground state was not performed in the dual picture but in a more optimised setting to capture fluctuations of the environment that act back onto the closed finite system of a graph. Here we stay in the dual picture because a modified coupling due to the environment is not obvious for excited states. With NLCEs one can obtain converging results for larger values of $h$ than with perturbation theory. As long as the correlation lengths are captured within the length scale of graphs considered it is reasonable to assume that NLCEs can converge. In contrast to perturbative expansions where order of perturbation and length scales are coupled, for NLCEs this is not the case any more since an exact calculation on a graph can be thought of as a resummation of an infinite order expansion on that graph. Consequently, the convergence properties of both approaches can be different.

With the NLCE applying our transformation $T^{\mathrm{pca}}$ we also calculated $\Delta$ and $\Delta_{\mathrm{bs}}$ using exact diagonalisations with ARPACK routines to obtain the low-energy spectrum and eigenvectors of $\mathcal{H}$. In Fig. 2 we show plots of the spin-flip gap for different numbers of vertices of the graphs used in the expansion and compare with extrapolations of the series results. The NLCE converges to values of $h \approx 0.5$ extending the convergence of the bare series. We also show Wynn extrapolations [60] with regard to the number of nodes of graphs in Fig. 2. Wynn extrapolations of a series $S_o$ depending on an expansion parameter $o$ are defined as

$$\frac{S_{o+1}S_{o-1} - S_o^2}{S_{o+1} - 2S_0 + S_{o-1}}. \tag{93}$$

These extrapolations extend the convergence of the NLCE a bit further but it still breaks down before the critical point at $h_c \approx 0.7610$ [4, 50]. One way to access critical exponents with NLCEs is to scale the spin-flip energy gap with respect to the number of vertices $N_v$ of graphs used in the expansion at the position $h_c \approx 0.7610$ of the estimated critical point. A logarithmic plot of this is shown in Fig. 3 together with a linear fit. This fit yielded an exponent of $\kappa = -0.51$. As in this model one would expect the gap to scale with the inverse correlation length this result implies that not the number of vertices $N_v$ but the square root of it scales in the same way as the correlation length. Although this analysis does not allow for a very precise determination of the critical point it clearly is consistent with a critical value of $h_c \approx 0.7610$ and hence shows that critical behaviour can be captured with NLCEs of excitation gaps.

The NLCE expansion of the bound-state gap converges up to $h \approx 0.35$. For a perturbative calculation of the bound-state energy it does not matter if one subtracts only the ground-state parts from the bound-state eigenvectors or both the ground-state and single-spin flip part as described in Eq. (70). Interestingly, the NLCE broke down earlier when only the ground-state part was subtracted so we always also subtracted the spin-flip part. Results are shown in Fig. 4. The reason for worse convergence in comparison to $\Delta$ is energetic overlap between bound states and the two-spin flip continuum [28]. This is a well known problem in all sorts of effective Hamiltonian theories and for example also shows up in quantum chemistry as intruder state problem on finite systems [61] or in graph-based continuous unitary transformations (gCUT) [62]. Only a finite number of eigenstates and eigenvectors exist in a finite system.

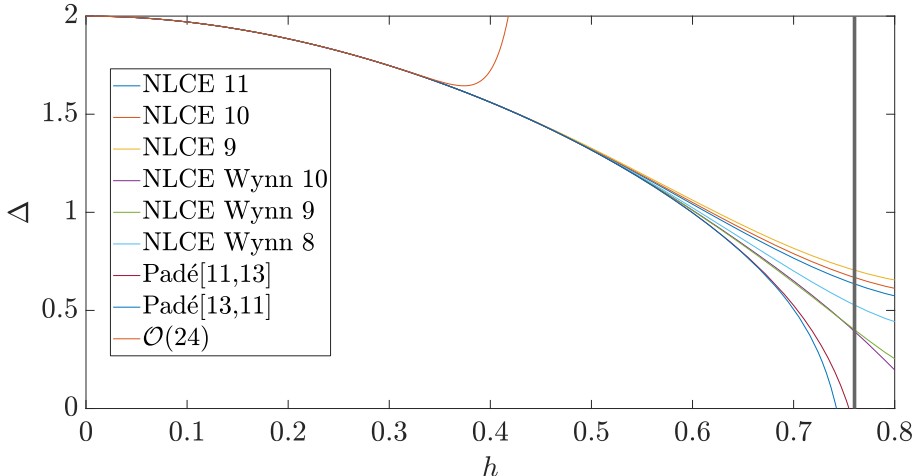

Figure 2: The figure shows an NLCE expansion of the spin-flip gap $\Delta$ in dependence of the number of vertices of the graphs taken into account. The expansion converges until around $h \approx 0.5$. The phase transition point $h_c \approx 0.7610$ [4, 50] is highlighted as a black vertical line. Wynn extrapolations of the NLCE expansion converge up to slightly larger values of $h$ but converge only slowly towards the critical point. Padé extrapolations are also shown together with the bare series.

Energetic overlap between two different sorts of formerly gapped quasi-particles shows up as an avoided level crossing. These avoided level crossings are also connected to exceptional points in the complex plane of the perturbation parameter that we follow adiabatically [63]. As pointed out in [61] either one follows adiabatically the low-lying state and loses transferability of the expansion or one tracks the right states but then has a problem of smoothness of the expansion around the avoided level crossing. A promising solution to overcome this problem was found in [62], where in the region of an avoided level crossing not exact but only approximate eigenstates were used to track the right diabatic states as well as possible and not the adiabatic ones any more. They used continuous unitary transformations based on the quasi-particle generator in Eq. (15) [1] but using a modified generator around the anti-level crossing. Next to observable characteristics they took a quantity known from the CORE method as characteristic to identify such pseudo-particle decay. For single-particle excitations not coupled to the ground state this quantity behaves similarly as the minimal eigenvalue of Eq. (71)

$$W_n = \sum_{m<n+1} P_m \sum_{m<n+1} \bar{P}_m \sum_{m<n+1} P_m.$$

While a generalization to the generic case seems not so clear within the CORE approach $W_n$ naturally shows up in our approach and can be used to identify particle-decay of higher energetic excitations or excitations coupled to the ground state. Indeed, Fig. 5 shows a graph where avoided level crossings related to the quasi-particle decay occur. As can be seen, the minimal eigenvalue $w_{\min}$ of Eq. (71) drops to zero as the two eigenvalues of the bound states and spin-flip states approach each other. While decay is expected for high-energy momentum modes in the thermodynamic limit the low-energy modes of the bound states are expected to remain stable. Hence, it could be possible to keep some decay channels open but to still do a linked-cluster expansion for the stable bound-state modes. A solution to this problem in our approach could be to not use exact projective eigenspaces around an avoided level crossing but only approximate eigenspaces in the spirit of [62], still demanding pairwise orthogonality of each space. A solution to this problem is beyond the scope of this paper. We stress that it is not clear if a parameter-free or even cluster-additive solution to this problem exists in general.

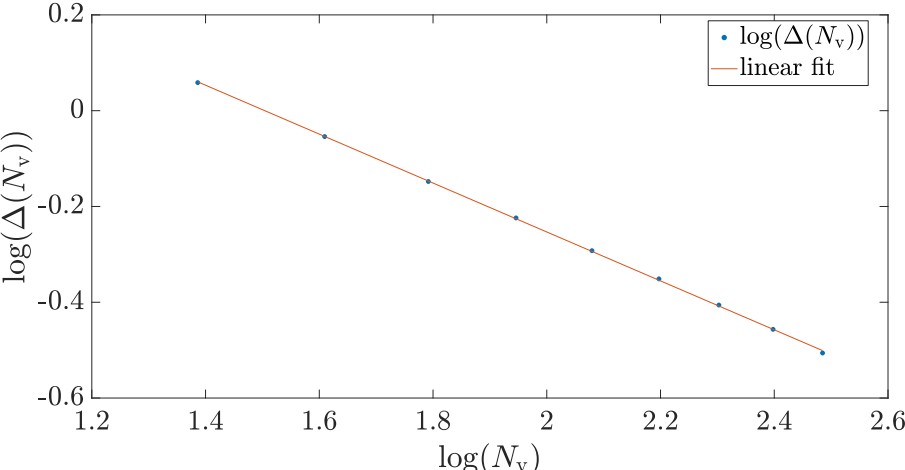

Figure 3: The plot shows the scaling of the energy gap $\Delta$ in the dependence of the maximum number of vertices $N_v$ of graphs used in the NLCE in a double-logarithmic plot. A linear fit of good quality shows that the behaviour is algebraic with an exponent of $\kappa = -0.51$.

# 5 Conclusions

We described how to construct a cluster-additive transformation for excitations of a Hamiltonian $\mathcal{H} = \mathcal{H}_0 + \lambda V$ with energies $e^n$ adiabatically connected to the energies $e_0^n$ of $\mathcal{H}_0$. The transformation only depends on the projectors of eigenspaces $e_0^m \leq e_0^n$ of $\mathcal{H}_0$ and the projectors of the adiabatically connected eigenspaces of $\mathcal{H}$. In that respect the transformation needs minimal information content compared to other genuine cluster-additive transformations while generalizing the well-known minimal transformation, which uses projectors on the eigenspace $e_0^n$ and the adiabatically connected space of $\mathcal{H}$ only, but is not cluster-additive in general. We also give the transformation explicitly in terms of projection operators, which implies basis independence and local expressibility of the perturbative expansion following from the projector expansion of Kato (25). As an application we performed a low-field linked-cluster expansion for spin-flip and two spin-flip bound state excitations in the transverse-field Ising model on the square lattice. We did this both perturbatively and non-perturbatively.

Both in the perturbative and non-perturbative setting the method is computationally very efficient. The complexity of perturbative calculations is similar to the TBOT method, which is the most efficient method for high-order matrix perturbation theory we know of. Non-perturbatively the complexity is that of Krylov-based diagonalisation methods. While perturbatively it is hard to come up with further improvements of the method, in non-perturbative applications using exact eigenvectors of finite-lattice Hamiltonians problems arising in the vicinity of avoided level crossings still present a major obstacle. Promising approaches to overcome this problem were given in [62]. To find a parameter-free and cluster-additive way of dealing with avoided-level crossings in the construction of effective Hamiltonians remains an important task for the future. If this is achieved the proposed transformation provides a highly efficient tool to perform linked-cluster expansions for excitations in generic Hamiltonians with the possibility to describe the decay of excitations accurately and efficiently.

We want to end the paper with possible applications of the introduced method. The minimal transformation only allows for a perturbative linked-cluster expansion of excitations that are

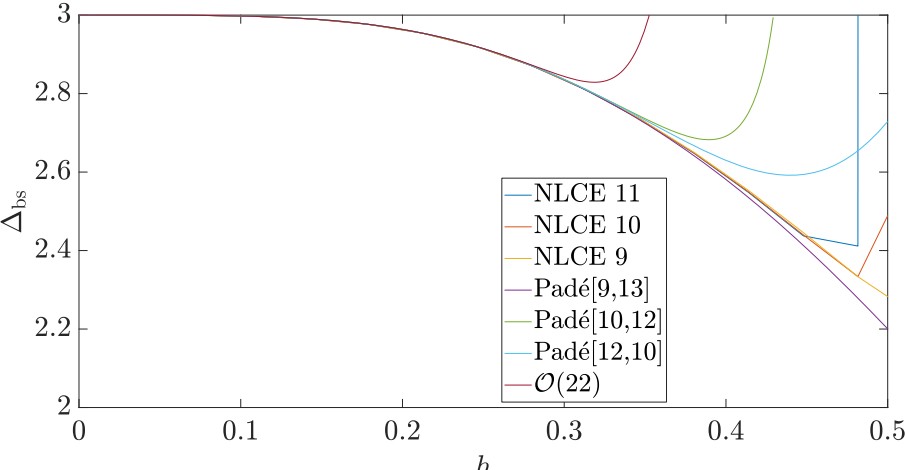

Figure 4: The figure shows an NLCE expansion of the bound-state gap $\Delta_{\mathrm{bs}}$ in dependence of the number of nodes of the graphs taken into account. The expansion converges only until around $h \approx 0.35$. The convergence problems are caused by avoided level crossings occurring on finite graphs. As more graphs are taken into account in the expansion convergence becomes gradually worse. Padé extrapolations and bare series results are also shown.

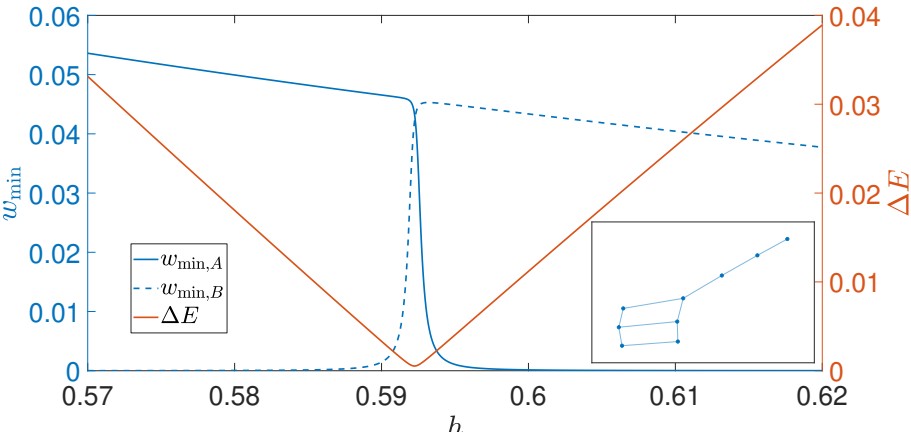

Figure 5: The figure shows the behaviour of the minimal eigenvalue $w_{\mathrm{min},A}$ of $W_2$ (blue line) in the vicinity of an avoided level crossing for the calculation of the effective Hamiltonian on a finite graph, which is plotted in the inset of the figure. In the same plot the energy difference $\Delta E$ between the lower end of the two-spin flip continuum and the maximum of the bound-state dispersion is plotted (red). One clearly recognizes that $w_{\mathrm{min},A}$ drops to a very small value as $\Delta E$ decreases. As a blue dashed line the minimal eigenvalue $w_{\mathrm{min},B}$ of a modified $W_2$ is shown, where one takes the formerly lower two-spin flip continuum state for the calculation of the bound-state effective Hamiltonian and rejects the state that was formerly the one with the highest energy of the bound states. The plot clearly suggests further away from the avoided level crossing the dashed blue curve would continue the solid blue one smoothly.

in a different symmetry sector than the ground state. In almost all low-field expansions this is not the case. While it is possible to perform such expansions with pCUT or MBOT these methods are less efficient than the method we propose. Hence, it promises to reach higher orders in low-field expansions in general, what we already showed specifically for the transverse-field

Ising model on the square lattice. High-field expansions of models where the ground state is coupled with the first excited states can also be computationally very demanding. An example is the Kitaev model in a field [64, 65]. The proposed transformation could help to reach higher orders for that system. Another advantage compared to pCUT is that we do not need an equidistant spectrum of $\mathcal{H}_0$. In [66] it was proposed to use the model-independent structure of the pCUT solution to treat systems with disorder or long-range interacting systems and this idea, coined white-graph expansion, was also successfully applied [10,12]. Using perturbative expansions of projectors we can do the same with this transformation but in a more general setting of non-equidistant $\mathcal{H}_0$. This can be utilized to perform white-graph expansions for the resolvent revealing the possibility of long-range low-field linked-cluster expansions and low-field linked-cluster expansions in the presence of quenched disorder.

# Acknowledgements

MH thanks Matthias Mühlhauser for fruitful discussions as well as for embeddings and graphs for the low-field expansion of the transverse-field Ising chain.

**Funding information** This work was funded by the Deutsche Forschungsgemeinschaft (DFG, German Research Foundation) - Project-ID 429529648 - TRR 306 QuCoLiMa (Quantum Cooperativity of Light and Matter). KPS and MH acknowledge the support by the Munich Quantum Valley, which is supported by the Bavarian state government with funds from the Hightech Agenda Bayern Plus.

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
