# Peer review of "Projective cluster-additive transformation for quantum lattice models"

_SciPost Physics, doi:SciPost Phys. 15, 097 (2023)_

## Round 1 · Referee Report · Anonymous (Referee 1) · 2023-7-7

Report

The authors have taken into account my remarks and accordingly modified the article. They notably added more details in the introduction on the method.

---

## Round 1 · Referee Report · Anonymous (Referee 2) · 2023-7-13

Report

The authors have addressed all of the referee's comments in the new manuscript, improving the readability and putting in context their advances.

---

## Round 1 · Author Response

Report 1
We thank the referee for thoroughly examining our work and raising several interesting points. We have revised in particular the introduction as recommended by the referee. This has helped to further improve the readability of our article. Below we address all specific points raised by the referee in his/her report:
Referee: “If the dimension of the Hilbert space is N (p4), the sum must go from 0 to N-1, or from 1 to N in Eqs. 4, 5, 6, ... “
Our answer: We thank the referee for this point. The Hilbert space dimension was wrong. We changed the dimension to a^N for a model with local Hilbert space dimension of a and N sites.
Referee: “If the dimension of the Hilbert space is really N (p4), then the eigenspaces H_O^n all have dimension 1, and H_0 is not only block-diagonal, but diagonal ?”
Our answer: Indeed, this would be the case if the Hilbert space dimension were N which is not the case since we address the general problem of degenerate blocks of arbitrary size.
Referee: “H^1_eff is the effective Hamitonian in the one particule space (p5). But according the the definition p.4, it is the block of the Hamilotian in the eigenspace of energy e1. Does it mean that all quasi particules have the same energy ? It's generally not the case (dispersion relation). Is it only the case for the unperturbed Hamiltonian H_0 (then it has to appear clearly) ?”
Our answer: We thank the referee for this point. Only for the unperturbed Hamiltonian the eigenstates all have the same energy in one block. Indeed, the distinction between energies of the unperturbed Hamiltonian and the full one was not clearly written. We improved this by adding the following two sentences on page 5:
“The block-diagonal form of $\mathcal{H}_{\mathrm{eff}}$ is specified by demanding that $\mathcal{H}_{\mathrm{eff}}^n$ contains the eigenstates adiabatically connected to the eigenstates of $\mathcal{H}_{0}^n$. The set of (possibly degenerate) energies of those eigenstates is denoted by $\boldsymbol{e}^n$. “
The unperturbed energies are always denoted with a subscript 0, whereas for the full Hamiltonian no subscript is used. We changed notation and use a bold e for the set of energies of the full Hamiltonian now to emphasize that it is not only one energy in each block but a set of different energies.
Referee: “ With all these unprecisions, it's hard to understand the meaning of Eq. 11, where two notations are simultaneously introduced. The |_1, and the overline.”
Our answer: We thank the referee for this point. In Eq. 11 it is not yet necessary to introduce the restriction |_1 and we therefore have erased this notion from Eq. 11. We have now just introduced this notation |_ in Eq. 13. We further added the extra sentence “In contrast to $\mathcal{H}_{\mathrm{eff}}^1$, $\bar{\mathcal{H}}_{\mathrm{eff}}^1$ is additive.” after Eq. 11.
Referee: “Between Eq. 16 and 17, what is X? (from the following, it seems to be a vector, but become clear only pages later)”
Our answer: We thank the referee for this point. This was a typographical error. It should be “As the first-order part of $\mathcal{S}$ has to decouple all blocks, it can be written as a sum of local operators.”
Referee: “There is a typo in Eq. 50 (H_A->H_B)”
Our answer: We thank the referee for this point. We changed the error in Eq. 49 (not 50).

Report 2
We thank the referee for his/her careful examination of our work and the overall recommendation for publication in SciPost Physics. In the revised version we have addressed the requested changes of the referee and we are convinced that this has helped to further improve the readability of our article, particularly of the introduction. Below we address all specific points raised by the referee in his/her report:
Referee: “- Even though the introduction is lengthy and very complete in a matter of references, many times concepts and quantities appear out of the blue. For example, the first time that "transformation" appears in the main text is on Page 3 in the sentence: "However, not every transformation yields an effective Hamiltonian that allows a decomposition of the form (3)". This is also the first time the concept of "effective Hamiltonian" appears. At this point, what kind of effective Hamiltonians are we talking about and what kind of transformations?”
Our answer: We thank the referee for this point. After equations 1 and 2 on the first page, where a splitting in H_0 and V is introduced, we added the sentences “Many times one is not interested in properties at all energies of the many-body Hamiltonian but only in the properties of the ground-state and a few low-lying excitations and thus in much less degrees of freedom. Conceptionally, one can try to find a transformation $T$ that maps the full Hamiltonian to an effective Hamiltonian $\mathcal{H}_{\mathrm{eff}}$ describing these relevant degrees of freedom only. In practice, in almost all cases one can not find this transformation exactly but has to resort to approximations.”.
Referee: “- Sentences are often a little confusing. For example, on Page 3: "In this paper we will introduce an optimal transformation: It shares the efficiency of the projective method, can also be applied non-perturbatively using the exact lowest eigenvectors and energies, but also allows for a cluster expansions with linked clusters only". What does "can also be applied" refer to if no previous application is mentioned? What contrast or addition is "but also" indicating at the end?”
Our answer: We thank the referee for the point. We have rewritten parts of this paragraph and changed the sentence (now on page 4) to “This method shares the efficiency of the projective method, can be applied non-perturbatively using the exact lowest eigenvectors and energies, and allows for cluster expansions with linked clusters only.”
Referee: “- Also, generally speaking, many methods are introduced and/or presented but usually, the discussion tends to be too technical. To anchor the methods for the non-expert reader, comments on the scope of validity of the methods are necessary. When are these methods exact? When do they give good qualitative or quantitative results? In which limit do they fail? Which model Hamiltonians can be solved? Which can not? In which dimensions? In which cases they are better than other standard methods? In which cases they are not?”
Our answer: We thank the referee for this point. At the beginning of the document after equation 1 on page 1 we added the sentence “We will assume throughout that $\mathcal{H}_0$ is solvable and has a gapped spectrum. ” and rephrase the next sentence as “The part $\mathcal{H}_0$ can therefore be written in diagonal form, …”. This is a requirement and a limitation. Assuming this, every model Hamiltonian can be treated. However, this does not mean results will be exact. On page 3 it is written “An obvious drawback of perturbative results is the limitation to the convergence radius of the perturbative expansion.”. This is an obvious drawback of any perturbative method. We added a sentence on page 2 to make clear that a linked-cluster expansion is essential in dimensions larger than one to obtain high-orders or large enough correlation lengths in NLCEs: “Especially in dimensions larger than one this becomes essential for obtaining high orders.“ Another sentence was added on page 3: “They are again only expected to converge within the quantum phase adiabatically connected to the limit $\lambda=0$.”. This should make clear that the expansion is only valid in one phase. Another sentence was added on page 3: “NLCEs have the potential to converge whenever a finite correlation length is present and to allow for scaling close to critical points.”. This should make clear that the crucial limitation is the correlation length. To compare with other methods the three sentences “In contrast to quantum Monte Carlo simulations, frustration poses no technical problem. NLCEs also do not suffer from high dimensions as density-matrix renormalization group does. The same holds true for the perturbative linked-cluster expansions.” were added on page 3. Let us further mention that in section 4 we discuss extensively the problem of avoided level crossings and that it is the biggest current problem of NLCEs. Further, also in section 5, we write that it is a future goal to find solutions for this problem.
Referee: “- I have worked before with Padé Approximants and have never observed spurious poles to be a source of big problems. Normally, spurious poles in the numerator and denominator cancel each other and the functions themselves behave continuously at such points. Small divergencies are observed only when the PAs are numerically evaluated and poles and zeroes do not match exactly. In your article, however, the coefficients are not exact and are only obtained up to a certain digit. The question: do the spurious poles arise because of the inexactitude of the coefficients? Or are they inherent in this kind of problem?”
Our answer: We thank the referee for this question. Spurious poles are inherent in this kind of problem. A similar conclusion was reached in previous publications J. Phys. A. Math. Gen. 24(12), 2863 (1991) and Phys. Rev. B - Condens. Matter Mater. Phys. 81(6) (2010), where the same series (to lower order) have been reached numerically with high precision or exactly with fractions.
Referee: “- Once the series is known, the Dlog Padé method can be used to obtain critical points and exponents. One calculates the logarithmic derivative of the series and then the PAs of the result. These PAs have poles and residues that give the values of the critical points and exponents. Typically, one can calculate all poles and corresponding residues for all the PAs [m,d] with m+d=n, and for several n. These points tend to form a curve of critical exponent vs critical point and are more concentrated around the real physical singularity. Have you made this analysis? Does it work in this case? Why is the PA [10,12] the only one mentioned on Page 18?”
Our answer: We thank the referee for this question. We have made an analysis of the families of PAs [m,n] of the logarithmic derivative of the series in the variable h^2 with m-n=d=const. To transform the series to a new series in h^2 is not uncommon practice. Unfortunately, we did not write this in the manuscript and falsely wrote the DLog-Pad\'{e} [10,12] extrapolant. We corrected the sentences to "As the series only contains even orders we made the analysis for the series in the variable $h^2$. Note that the maximum order of the series in this variable is $12$. We found that only the families with $d=\pm 1$ show converging behaviour and that the family $d=1$ appears to be better converged. For the $d=-1$ family the extrapolation of the highest order, i.e. the $[6,5]$ DLog-Pad\'{e} extrapolant, yields a critical point $h_c=0.727$ and a critical exponent $\nu=0.417$. From the highest-order $[5,6]$ DLog-Pad\'{e} extrapolant of the better-converged $d=1$ family one obtains a critical point $h_c=0.762$ and a critical exponent $\nu=0.649$." For the one-particle gap, only two families show non-spurious behaviour: d=-1 and d=1. We have now also added the representative of the d=-1 family of highest order in the manuscript. The obtained series for the problem at hand can therefore not be extrapolated in a fully satisfactory fashion due to the spurious poles, also not when keeping the series in its original form as a series of h. The situation is even worse for the two-particle bound state. However, we want to stress that the focus of this work and this example is to introduce our new projective transformation and to demonstrate its optimal properties, which have nothing to do with the behaviour of these two specific perturbative series.
Referee: “A little typo: at some point, the article reads "lwolwo-field ordered phase"
Our answer: We thank the referee for this point. We changed „lwo-field ordered phase" to "low-field ordered phase".
Referee: “Some little things like the name of Section 4 need to be corrected: "Low-field expansion for transverse-field Ising model on square lattice". Things are missing. Should be something like "Low-field expansion for the transverse-field Ising model on the/a square lattice".”
Our answer: We thank the referee for this point. We changed the title of section 4 from "Low-field expansion for transverse-field Ising model on square lattice" to "Low-field expansion for the transverse-field Ising model on the square lattice".
Throughout the article we made corrections of typos and grammar as asked for by the referee.

---

## Round 1 · List of Changes

• The Hilbert space dimension was wrong. We changed the dimension to a^N for a model with local Hilbert space dimension of a and N sites.
  • We added the following two sentences on page 5: “The block-diagonal form of $\mathcal{H}_{\mathrm{eff}}$ is specified by demanding that $\mathcal{H}_{\mathrm{eff}}^n$ contains the eigenstates adiabatically connected to the eigenstates of $\mathcal{H}_{0}^n$. The set of (possibly degenerate) energies of those eigenstates is denoted by $\boldsymbol{e}^n$. “
  • We changed notation and use a bold e for the set of energies of the full Hamiltonian now to emphasize that it is not only one energy in each block but a set of different energies.
  • In Eq. 11 we have erased the notion |1. We have now just introduced this notation | in Eq. 13.
  • We further added the extra sentence “In contrast to $\mathcal{H}_{\mathrm{eff}}^1$, $\bar{\mathcal{H}}_{\mathrm{eff}}^1$ is additive.” after Eq. 11.
  • We changed a sentence between Eq. 16 and 17 to: “As the first-order part of $\mathcal{S}$ has to decouple all blocks, it can be written as a sum of local operators.”
  • We changed the error in Eq. 49 (not 50).
  • After equations 1 and 2 on the first page, where a splitting in H_0 and V is introduced, we added the sentences “Many times one is not interested in properties at all energies of the many-body Hamiltonian but only in the properties of the ground-state and a few low-lying excitations and thus in much less degrees of freedom. Conceptionally, one can try to find a transformation $T$ that maps the full Hamiltonian to an effective Hamiltonian $\mathcal{H}_{\mathrm{eff}}$ describing these relevant degrees of freedom only. In practice, in almost all cases one can not find this transformation exactly but has to resort to approximations.”.
  • We have rewritten parts of a paragraph on page 3 and changed one sentence (now on page 4) to “This method shares the efficiency of the projective method, can be applied non-perturbatively using the exact lowest eigenvectors and energies, and allows for cluster expansions with linked clusters only.”
  • At the beginning of the document after equation 1 on page 1 we added the sentence “We will assume throughout that $\mathcal{H}_0$ is solvable and has a gapped spectrum. ” and rephrase the next sentence as “The part $\mathcal{H}_0$ can therefore be written in diagonal form, …”.
  • We added a sentence on page 2 to make clear that a linked-cluster expansion is essential in dimensions larger than one to obtain high-orders or large enough correlation lengths in NLCEs: “Especially in dimensions larger than one this becomes essential for obtaining high orders.“ Another sentence was added on page 3: “They are again only expected to converge within the quantum phase adiabatically connected to the limit $\lambda=0$.”.
  • Another sentence was added on page 3: “NLCEs have the potential to converge whenever a finite correlation length is present and to allow for scaling close to critical points.”.
  • To compare with other methods the three sentences “In contrast to quantum Monte Carlo simulations, frustration poses no technical problem. NLCEs also do not suffer from high dimensions as density-matrix renormalization group does. The same holds true for the perturbative linked-cluster expansions.” were added on page 3.
  • We corrected sentences on page 17 and 18 to "As the series only contains even orders we made the analysis for the series in the variable $h^2$. Note that the maximum order of the series in this variable is $12$. We found that only the families with $d=\pm 1$ show converging behaviour and that the family $d=1$ appears to be better converged. For the $d=-1$ family the extrapolation of the highest order, i.e. the $[6,5]$ DLog-Pad\'{e} extrapolant, yields a critical point $h_c=0.727$ and a critical exponent $\nu=0.417$. From the highest-order $[5,6]$ DLog-Pad\'{e} extrapolant of the better-converged $d=1$ family one obtains a critical point $h_c=0.762$ and a critical exponent $\nu=0.649$."

---

## Editorial Decision

published